# A standard convention for particle-level Monte Carlo event-variation weights

**The MCnet Community**

Editors: Enrico Bothmann[1], Andy Buckley[2], Christian Gütschow[3],
Stefan Prestel[4], Marek Schönherr[5] and Peter Skands[6]

Supporters: Jeppe R. Andersen[5], Saptaparna Bhattacharya[7], Jonathan Butterworth[3],
Gurpreet Singh Chahal[5], Louie Corpe[8], Leif Gellersen[6], Matthew Gignac[9], Stefan
Höche[10], Deepak Kar[11], Frank Krauss[5], Jan Kretzschmar[12], Leif Lönnblad[4],
Josh McFayden[13], Andreas Papaefstathiou[14], Simon Plätzer[15], Steffen Schumann[1],
Michael H. Seymour[16], Frank Siegert[17] and Andrzej Siódmok[18]

**1** Institut für Theoretische Physik, Georg-August-Universität, Göttingen, Germany
**2** School of Physics & Astronomy, University of Glasgow, Glasgow, UK
**3** Department of Physics & Astronomy, University College London, London, UK
**4** Department of Astronomy & Theoretical Physics, Lund University, Lund, Sweden
**5** Institute for Particle Physics Phenomenology, Durham University, UK
**6** School of Physics & Astronomy, Monash University, Clayton VIC 3800, Australia
**7** Department of Physics and Astronomy, Northwestern University, Evanston IL, USA
**8** CERN, Meyrin, Switzerland
**9** University of California Santa Cruz, Santa Cruz CA, USA
**10** Fermi National Accelerator Laboratory, Batavia, IL 60510, USA
**11** University of Witwatersrand, Johannesburg, South Africa
**12** University of Liverpool, Liverpool, UK
**13** Physics and Astronomy Department, University of Sussex, Brighton, UK
**14** Department of Physics, Kennesaw State University, Kennesaw, GA 30144, U.S.A.
**15** Institute of Physics, University of Graz, 8010 Graz, Austria
**16** University of Manchester, Manchester, UK
**17** Institut für Kern- und Teilchenphysik, TU Dresden, Dresden, Germany
**18** Jagiellonian University, Cracow, Poland

## Abstract

Streams of event weights in particle-level Monte Carlo event generators are a convenient and immensely CPU-efficient approach to express systematic uncertainties in phenomenology calculations, providing systematic variations on the nominal prediction within a single event sample. But the lack of a common standard for labelling these variation streams across different tools has proven to be a major limitation for event-processing tools and analysers alike. Here we propose a well-defined, extensible community standard for the naming, ordering, and interpretation of weight streams that will serve as the basis for semantically correct parsing and combination of such variations in both theoretical and experimental studies.

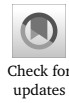

# 1 Introduction

With the inexorable rise in CPU cost of high-precision Monte Carlo (MC) collision-event samples [1–3], and the experimental need for huge volumes of such samples, reweighting as opposed to explicit re-running of MC event samples has become the norm. In this approach, a systematic variation to the *nominal* MC generator configuration — the one used to perform the original sampling of event properties — can be obtained without regeneration by identifying the ratio of probability density between the variation and the nominal for each event. In MC programs this ratio is often easily calculable by rapid re-evaluation of a matrix-element or parton-shower kernel formula. The ratio factor is multiplied on to the nominal-configuration event weight (resulting from weighted or kinematically biased generation strategies [4]) to give the event-specific *variation weight*.

Events which are more likely under the variation than the nominal assumptions have a variation weight greater than the nominal weight, and *vice versa*. Provided each systematic variation's phase-space distribution has sufficient overlap with the nominal distribution from which the events are sampled (i.e. that the two distributions have common support), using the variation weight in place of the nominal one for histogramming or other physics applications with the single event sample produces observable distributions asymptotically equal to those from explicit generation from the variation model. In practice, weighting dilutes the statistical power of an MC event sample, but for mild variations and for uncertainty estimation purposes, the convenience, speed, and reduced storage requirements make this compromise acceptable.

Including hard-process scale variations, PDF error sets, variations in parton-shower tunes, electroweak corrections, heavy-flavour schemes, BSM physics, and more besides, in current usage it is not uncommon for hundreds of physics variations to be found in each MC event, stored in its *weight vector*[1] [5–23].

However, the character-string names used to identify these vectors' elements — the weight elements corresponding to one variation through an event dataset being called a *weight stream* and having a unique name — are not yet standardised between generators, leading to complexity, duplicated decoding attempts, and inevitably errors in analysis codes within experimental collaborations and in public MC tools such as Rivet [24] and MadAnalysis [25]. Previous attempts [26] have been made to standardise the names and values in weight streams for the parton-level LHE format [27], and are heavily used. However, the connected attempt to standardise the propagation of such event weights to particle-level, i.e. fully exclusive events in the absence of detector interactions and primarily in the HepMC format [28], have not in practice seen sufficient uptake and a variety of *de facto* naming formats are in circulation via different generator codes. This situation causes problems and effort-duplication for the many direct and indirect users of particle-level events.

In this document we address these issues with a new, more complete, extensible, and formally defined format for particle-level weight-stream naming, in particular for event generators producing the HepMC format and for users of such events. We additionally place essential requirements on the ordering of such variations in generator weight-vectors, for efficient and reproducible downstream processing, and codify the semantic meaning of the variation-weight values. This summarises the consensus from a year-long discussion process among representatives of the main ME+shower+hadronisation event generator teams, which combines the best features of the various preceding *ad hoc* schemes, and defines a new community standard for LHC Run 3.

---

[1]An ordered tuple, rather than a physics vector in the sense of vector spaces.

## 2  Particle-level weight-name structure

The main role of event weights is to indicate the shifts of differential cross-sections due to variations in the physics model — from alternative hard-process scales and PDFs, to parton shower options, heavy-flavour treatments, electroweak corrections, and so on. Simultaneous combinations of such elementary variations are also commonplace. The component variations hence need to be clearly identifiable from the names of the weight streams.

In HepMC these stream-names are simple character strings: there is no more complex data-structure available, and hence particle-level variations need to be encoded in a "flat" form, with each stream treated as independent, i.e. there is no opportunity for "grouping" of streams or "inheritance" of common variations into subsets of streams. In this proposal we hence describe a scheme for structuring of the available string to specify in a uniform way the parameters varied in each stream. This notably does not specify exact rules for combination, although in many standard cases these are well-established: we leave this possibility, which requires also detailed discussion and perhaps updating of the LHE event format, as a possible extension to be built on the foundation of this particle-level standard.

Our agreed general form for particle-level weight names is the following: either a special nominal weight name, preferably the empty string but with a few acceptable alternatives, or a variation weight name, in the format

```
PREFIX__LEVEL.KEY1=VALUE__LEVEL.KEY2=VALUE...
```

i.e. a prefix and a non-empty set of `key=value` blocks separated by double-underscores. The double-underscore serves the dual purpose of both allowing single underscores to appear in block components, and of more clearly visually distinguishing the blocks; both features aid human readability. The keys may have multiple levels, separated by dots. The prefix and the `=VALUE` part of each block are optional, and the prefix and blocks obviously cannot themselves contain the double-underscore separator. A practical example could be

```
MUR=1.0__MUF=2.0__PDF=123456__CKKW.MUQ=20 .
```

A more formal specification of weight names in general may hence be given in W3C EBNF [29] notation:

```
WEIGHTNAME ::= NOMNAME | VARNAME
NOMNAME ::= "" | "NOMINAL" | "DEFAULT" | "WEIGHT" | "0"
VARNAME ::= (PREFIX "__")? BLOCK ("__" BLOCK)*
PREFIX ::= "USER" | "AUX" | "EXTRA" | "IRREG"
BLOCK ::= KEY ("=" VALUE)?
KEY ::= IDENT ("." IDENT)* - NOMNAME
VALUE ::= IDENT | NUM
IDENT ::= ALPHA (ALPHA | DIGIT | "_")* - "__"
NUM ::= ("+" | "-")? DIGIT+ ("." DIGIT+)? EXPO?
EXPO ::= ("e" | "E") ("+" | "-")? DIGIT+
DIGIT ::= [0-9]
ALPHA ::= [A-Za-z]
```

where `NOMNAME` represents the special-case name for the default or nominal weight stream, to be discussed in Section 2.2. Analysis tools must always perform a case-insensitive string comparison, as all conceivable case-conventions already exist in established MC generator outputs.

The context given, we now explain in turn the allowed characters, the nominal weight name, the role and allowed values of the variation prefix, and the formatting principles and predefined types of variation key/value block.

## 2.1  Allowed characters

For both nominal and variation weights, there is little argument for use of a character set beyond the ASCII set, especially as extended "wide" character sets are problematic in many older or numerically focused languages and it is simplest to avoid issues around use of extended character encodings. All weight names are hence restricted to the basic alphanumeric characters and the following minimal set of punctuation characters:

- equal sign: =

- underscore: _

- full stop: .

- plus: +

- minus: –

where the plus and minus symbols are only allowed as a sign for numerical values or in exponentials, but not in level or key names, e.g. as a hyphen.

This set of characters has been arrived at by consideration of several factors that constrain a free-for-all approach:

- Restricting the set to core Unix and bash file-name characters should minimise the need for escaping, quoting or character translation in shell commands or file names. This excludes characters such as quote marks, $, >, <, or ; , which have special meaning in many scripting languages.

- For familiarity and for ease of translation to and from executable-code forms, the usual programming language restrictions on identifier names to be made purely of alphanumeric characters apply, including a prohibition on starting an identifier name with a digit.

- Weight names are often placed in brackets or parentheses of some sort to indicate that they are variations — including bracketing characters in the weight names themselves would therefore cause parsing difficulties that are best avoided.

- White-space characters are not permitted as they can be a source of headaches for the user. The weight name is a key in a C++ map and so "Name", " Name " (outer padding using a single white-space character on each side) and "  Name  " (outer padding using two white-space characters on each side) would be interpreted as distinct keys.

- In particular, newline and tab characters in a weight name lead to confusion when printed, are difficult to distinguish from (also discouraged) spaces, and can cause parsing difficulties in text-based formats.

- Comma or colon characters are not allowed, as they are natural separators or indicators of hierarchies, and hence introduce confusion with the double-underscore and dot syntax used for those concepts. Blocking their use in weight names means they remain available for use as separator characters in higher-level applications which need to handle weight names.

We also note that the minus, plus, and equals symbols and potentially other "operators" can cause issues when trying to save branches in ROOT [30]. However, this is a reflection of one format's limitation, and when automatically handling numerical variations minus signs are a natural occurrence both in the radix and exponent. Translators of event-weighted data

to ROOT-based data formats (and any other base format with such naming sensitivity) are advised to explicitly perform a translation between literal plus and minus signs and a ROOT-safe alphanumeric encoding of them, e.g. `-1.0` → `_MINUS_1.0`, and if necessary similar replacement of the equals-sign key/value separators with text or a ROOT-safe symbol.

## 2.2 The nominal/default weight

While several MC-generator variation modes are of *a priori* equal physics validity, particularly between the central members of several PDF global fits, a run of simulated events always has a nominal or default configuration: the one from which the matrix element and parton shower phase-spaces were rejection-sampled. This is typically unweighted, and hence the events will have uniform or near-uniform rejection sampling weights in the nominal stream. The other — variation — weight streams then re-evaluate the kinematics, mostly without the possibility of rejection, to obtain a modified weight: as this modification depends on the full event phase-space, each event's variation weights will deviate from uniformity in a unique way.

Without making a judgement about the physicality of particular configurations, although the nominal configuration is almost always a sensible one, it is extremely useful for an MC sample to have an easily identifiable default weight. It is most obvious that this should correspond to the nominal configuration used for the original sampling. Even for setups with multiple equivalent candidates for the default weight, one of them should be assigned default status, rather than forcing decoding on the user even for the simplest visualisation.

The nominal weight name should be easily identifiable even when parsing a list of jumbled up weight names. Recognising the range of established conventions in long-public generator codes, a set of acceptable names is required: these are `NOMINAL`, `DEFAULT`, `WEIGHT`, the digit `0`, or the empty string. Exactly one instance from this set must appear in a given list of weight-vector names.

## 2.3 Variation prefixes

The optional prefix and its delimiting double-underscore are for the purpose of discriminating physically meaningful weight streams from debug or bookkeeping ones, and for distinguishing between "official" and user-defined physical weights.

Much as every event-record graph contains elements intended more for debugging use by MC authors than as physically-meaningful event history for analysers, it is natural that event generator authors wish to encode debugging elements into their events' weight vectors. These, however, must appear in an unambiguous form which can be automatically skipped (at least by default) in analysis tools.

In discussion, it has been identified that there are at least two reasons to prefix weights with a "class" marker: one is this hiding of truly auxiliary, unphysical weight streams, and the other is to distinguish between physical weight streams "blessed" by the generator authors, and those which are physical but unblessed: "user" weights. The following prefixes may be used to classify such auxiliary weights:

**No prefix:** Canonical cross-section variations: good to process and plot (e.g. standard scale and PDF variations, for which a standard naming convention is given in the following section). As many weights as possible should come in this minimal form.

`USER, AUX:` Optional indication of non-standard cross-section variations: good to process and plot (e.g. user-defined shower variations). This prefix exists only to indicate the "unblessed" nature of the variations from the perspective of the generator authors: it should be treated equivalently to the no-prefix form for observable calculations by downstream tools.

`EXTRA`: Supplementary information, exists for every event, but either not a cross-section variation (e.g. Sherpa's `NTrials`) or requiring further manipulation to obtain a weight variation that can be used equivalently to the nominal weight. They are hence not required to behave like "physical" weight variations, and are not to be plotted or otherwise used as a physical variation by default.

`IRREG`: Irregular/generator specific: weights of this type are not processed by default (e.g. fixed-order information). This type of variation weight should always be appended at the end of the weight vector, such that it's easier to skip the `IRREG`-type group of variation weights (see Section 3).

While the choice between no-prefix and a `USER/AUX` prefix is largely a matter of taste, correctly indicating `EXTRA` and `IRREG` weight streams is mandatory, to ensure physically correct processing of events.

## 2.4 Variation blocks

Each block should represent one distinguishing characteristic of the weight stream, with the full set of blocks fully specifying how the stream deviates from the nominal configuration. The expectation is that any implied run settings not listed take the values used by the nominal stream, though null-variations such as scale multipliers of ×1.0 are allowed to be included if desired, at the cost of a more lengthy weight name, whose physics purpose is less immediately obvious to human readers.

It consists of a key, which can contain several dot-delimited levels, and an optional value appended with a delimiting equal sign to the right-hand side of the key. Neither the key nor the value can contain a double-underscore as this is the block delimiter, but single underscores are permitted. If the equals-sign and RHS value are omitted, the key will be interpreted as equal to a boolean "true" value.

Several standard key structures are established by existing practice and common use-cases, with an eye to coherent extension of the scheme as MC generator mechanisms evolve in sophistication and ability to use variation weights:

- For PDF variations, the most common scheme will be to specify a single PDF set, either for both beams or for the single composite-particle beam. In this case the key should either be `LHAPDF`, to specify that the following value is the LHAPDF set ID [31] and can be used for automatic uncertainty-band construction by standard analysis tools; or a custom key ending in `PDF` — e.g. `PY8PDF` — to specify a PDF identifier specific to the generator or application. In the latter case, it is recommended that the generator document the custom key in the event metadata, e.g. in the HepMC `GenEvent` or `GenRun` objects' custom attributes.

- For use-cases where multiple composite beams are in use at the same time, there may be a need to specify separate PDFs and factorization scales for each beam. In this case, the `*PDF` and `MUF` block names can be followed by an extra level specifying which beam it refers to, e.g. `LHAPDF.BEAM1` or `MUF.BEAM2`. The versions described above without these qualifying beam identifiers are to be interpreted as a compact syntax for simultaneously setting equal values for both beams.

- For scale variations, similar structures using `MUR` and `MUF` keys (without underscores) followed by scale-modifying multiplicative factors for renormalization and factorization scale shifts should be used. These are easily extendable to other types of scale variations, but should always include `MU`. In general, the scaling given should be linear in the energy

scale, not the squared form. However, a squared scale (e.g. `MUR2`) may be specified to avoid square roots in the token values.

- As well as SM parameters in the calculation, variations to encode changes in BSM-model parameters are a major use-case of event weights. There are too many models to pre-specify standard schemes for them all. They should, however, follow the general syntax described here, and as consistency aids cooperation we advise that model developers and users coordinate to match naming schemes as far as possible, e.g. based on the names and parameters in established UFO packages.

In this context we can see that the variation name

```
MUR=0.5__MUF=2.0__PDF=42
```

represents scalings down and up respectively of the renormalization and factorization scales by factors of two, and use of the internal PDF number 42. As there are no distinguishing levels to these keys, they may be assumed to apply to all parts of the MC generation. Introducing such levels we might find, e.g.

```
ME.MUR=0.5__ME.MUF=2.0__LHAPDF=123456__CKKW.MUQ=20
```

where the scale variations apply only to the matrix element (ME), a new merging scale `MUQ` has been introduced to the CKKW jet-merging component, and the PDF ID is now the global LHAPDF ID code rather than a run-specific index. Further distinctions such as application of different scalings to different parton showers can be applied, e.g.

```
ME.MUR=1.0__ME.MUF=2.0__ME.LHAPDF=123456__ISR.MYPDF=14
    __ISR.MUR=1.0__ISR.MUF=2.0__FSR.MUR2=2__FSR.MUR=1.0
    __CKKW.MUQ=20
```

The keys used in variation blocks can be different from the exact forms used in these examples, although certain weight name components are fairly well established already, and uniformity is a virtue for users if there is no compelling reason to deviate. Values corresponding to absolute energy scales shall be given in GeV.

Ideally, attention should be paid to how floating point values are represented, e.g. `MUR=1.000000E+00__MUF=2.000000E+00` is unnecessarily precise and impacts legibility. However, this formatting control may be technically limited by implementation language, and parsers should expect to recognise numeric tokens in a variety of standard forms. More extensive descriptions of the functional forms of scales as well as weight combination prescriptions can be added to the run meta if required.

## 3 Weight-vector selection and ordering

It has long been standard for the first weight in the vector to be the nominal, before weight names became common. This ordering has been broken in recent years by some generators using weight names, not helped by a bug in the HepMC ASCII-format writer which re-sorted output names (but not their values!) into alphabetical order.[2] While not an absolute requirement if using the standardised nominal-stream names, we strongly recommend that the nominal weight be placed first in the weight vector.

For analysis tools, the selection of weight subsets is typically very important. The benefit of weighting is its great speed with respect to explicit re-generation with a different nominal

---

[2]This bug was fixed in HepMC 2.06.11 and should no longer be considered a viable excuse for not paying attention to ordering.

configuration. But having $\mathcal{O}(1000)$ weights in a Monte Carlo setup adds often-unwanted analysis overhead for users who do not always want to analyse all variations for every iteration of their analysis. For example, the Rivet analysis toolkit permits both positive- and negative-condition filtering of weight streams based on numeric-index and regular-expression matching. As string processing is computationally slow and current generators use a stable ordering of weight streams in the vector, this matching is performed only once: a list of the desired weight-subset's numerical indices is obtained from the first event, and used directly from then on. The alternative of checking thousands of filter conditions for every event in case the weight-vector ordering had changed would be prohibitively slow. This established behaviour and opportunity for substantial performance optimisations in event processing means it is important to codify this ordering stability:

- the number of physical weight streams and their weight-vector positions must not change from event to event.

The existence of auxiliary weights, and of complex setups such as correlated NLO sub-events (where the set of weight names can differ event-to-event, as each event type is produced in an independent code block) can lead to situations where the maximal length of the weight vector is often only known at the end of the run. However, these latter weight streams are typically also auxiliary and not intended for analysis. Due to the "first event" inference of physical weight locations, in such cases it is important to structure the weight vector in such a way that the physical variation names for physics uses (and hence which need to be known right from the start) come first:

- Auxiliary weights that might not even exist for every event must be appended at the end such that only the unphysical tail of the weight vector changes in length and/or ordering.

## 4 Weight manipulation

Finally, a note on the meaning of weight values:

- Standard weights (i.e. those with empty, USER, or AUX prefixes) must all have equivalent semantic meaning to the nominal weight.[3]

This rule forbids use of partially-meaningful physical weight streams which need to be combined with the nominal or with other variation weights in order to be used for physics predictions. Any weight-combining logic of this sort should happen inside the Monte Carlo event generators and be hidden from the user. For example:

- if a setup contains a default prediction and renormalisation-scale variations, the scale-variation weight should correspond to the final cross-section contribution (apart from global normalisation factors);

- if a setup contains a default prediction and a weight that allows to filter out a subset of the events, satisfying certain selection properties, the filter weight should be equal to the nominal weight if the event is to be accepted or zero otherwise.

Of course, this does not mean that sets of variations, e.g. for PDF uncertainties, should be subjected to a combination attempt prior to the generator writing out the event, provided each variation in the set corresponds to a standalone physical variation of the event prediction.

---

[3]The careful wording here reflects that the nominal stream may not represent an entire physical event, cf. correlated counter-events in higher-order MC simulations. But weight streams are orthogonal to this: they must not introduce new semantic meanings to the event over those already in use for the nominal.

# 5 Summary

Variation weight-streams for MC-generator events are an increasingly important tool for propagating systematic uncertainties in the phenomenology calculations into observables for both theoretical and experimental studies. A major limitation on this usefulness is the lack of a common standard for labelling — and hence semantically correct processing and combination — of the variation streams. In this document we have proposed a well-defined base standard for such naming, including a general syntax and a minimal set of agreed names for the most common variation concepts. This format has been agreed by the signed MC-generator authors and users, and will serve as the basis for more portable event processing, and further community standards.

# Acknowledgements

This work was supported in part by the European Union as part of the Marie Sklodowska-Curie Innovative Training Network MCnetITN3 (grant agreement no. 722104). Of the editors, AB thanks the Royal Society for University Research Fellowship grant UF160548; AB and CG thank STFC for funding through the UK experimental particle physics consolidated grants programme; CG also thanks STFC for supporting the SWIFT-HEP project (grant ST/V002627/1); SP would like to acknowledge funding from the Swedish Research Council (Vetenskapsrådet), contract numbers 2016-05996 and 2020-04303; MS is funded by the Royal Society through a University Research Fellowship (URF\R1\180549) and an Enhancement Award (RGF\EA\181033 and CEC19\100349), and supported by the STFC under grant agreement ST/P001246/1.

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
