# Peer review of "A standard convention for particle-level Monte Carlo event-variation weights"

_SciPost Physics Core, doi:SciPost Phys. Core 6, 007 (2023)_

## Round 2 · Referee Report · Anonymous (Referee 1) · 2022-4-29

Report

The paper proposes a standard for naming and organizing event metadata ("weights") in Monte-Carlo generator and analysis programs.

The paper is not a physics paper. It is intended to help with bookkeeping issues which can easily become a significant bottleneck in maintaining and using HEP software. Therefore, a successful effort to standardize conventions can be very useful, and a document such as the present one should be suitable for publication in a major journal despite the lack of proper physics content.

In the past, there have been various attempts at standards in HEP, most notably several "Les Houches Accords". Since they were agreed upon by major players in their respective areas of research, they were usually adopted by the community and are still in use many years later. However, it should also be noted that with the evolvement and shifting focus of the field, formal conventions tend to persist in environments which have not been anticipated when they were first designed. Applications of previous "standards" often modify or plainly contradict the intention of the original setup, causing new problems for the practitioner. Many of these issues can be avoided if in the original document, all relevant content is stated as precisely as possible and in formal terms.

Unfortunately, while the intentions of the submitted document are evident, its current form falls short of this goal on various levels.

Major issues:

  1. The document is rather vague about its intended audience. It has been submitted by the MCnet community, and it reflects on upcoming simulation runs for LHC Run 3. It refers to software tools such as the HepMC event format that are currently employed by some part of the HEP community. If the intended audience was indeed restricted to this community and time scale, the proper form of the paper would be a public MCnet note; in this case I would not see justification for a journal publication.

Conversely, if the intended audience was larger, the authors should (1) state in concrete terms which API properties and version of currently existing software are relevant for the implementation of the intended standard, and (2) relate to a larger environment of Monte-Carlo simulation for current and future experiments, e.g., how it should apply to other existing or foreseen software frameworks, beyond the LHC, and to expected future developments such as NNLO-level simulation tools. I think that a standard for indexing MC event metadata could become very useful for the whole community.

  1. Basic terms such as 'weight', 'event', 'particle', 'stream', 'vector', etc. must be defined at the beginning of the document in self-contained and formal terms as far as they are relevant. The technical definition must be abstracted from the physical interpretation. In the current text, they are introduced in a rather casual way, assuming working knowledge (e.g., with HepMC) that is not evident for users not familiar with 2022 LHC internals. The terms should be used consistently throughout the document.

  2. The scope of the standard must be stated. I understand that the concrete syntax only defines the format of key identifiers in a lookup table. However, there are also various recommendations for the structure of this table (per event record), and there are remarks concerning the complete stream of records. This leaves any reader confused about which data objects are actually covered by the standard.

  3. It has to be stated whether and how the standard applies to data independent of their particular storage structure. At the end of the document there is a paragraph on optimization (i.e., using numeric indices and a global lookup table mixed with event-local keys). It is not obvious how such optimization interacts with the syntax and semantics introduced at the beginning, or whether it actually contradicts the normative part.

General remarks:

A data-format standard serves three purposes. A writer (human or computer) is enabled to construct a conforming data structure, a parser is enabled to correctly process the data syntax, and an interpreter is enabled to process the semantics as intended.

To this end, the standard definition must be written such that a human or computer can unambiguously decide whether a given data set is standard-conforming or not. The current draft is full of unclear terms like 'may', or 'should'; I do not see how it actually defines anything.

I recommend that the document clearly separates its formal part, namely the definitions of scope, terms, syntax and semantics, from informal additions. The formal part alone allows for writing a correct programming interface. The informal part may include examples, context, reasons, use cases, or best-practice recommendations. Actually, worked-out examples would be rather useful and could help to straighten the core of the document.

Some details:

  • Sec. 2: The term 'particle-level' is misleading without context -- is the weight pertaining to a single electron? Since the detector simulation environment is not referred to in the standard, I recommend to use predefined technical data terms where possible.

  • Syntax: it is not stated whether character case is meaningful or not, i.e., whether "NAME" is different from "Name". It is not stated whether keywords such as "NOMINAL" must be upper case. (Actually, at the end of 2.2. there is a remark which suggests that the syntax might be case-insensitive, but this is difficult to spot, and it is again just a 'should' clause.) It might be necessary to specify the character encoding. This depends on whether the standard refers to abstract names or to their representation in storage, which is not evident in the present text.

  • Syntax: it is not clear without reading the text several times what the allowed characters for ALPHA actually are. To avoid all ambiguities, directly define ALPHA as a character set at the point where it appears. (Same for DIGIT).

  • Syntax: the 'usual standard form' of an integer or floating-point number is unclear. The writer of a parser, for instance, must exactly know which formats to allow and accept. Regarding floating-point, it must be stated what the allowed ranges are and whether numeric or textual reading/comparison is to be applied (cf. numbers differing in the 31st decimal), whether +0.0 is equal to -0.0, etc. All of this is relevant if the key strings themselves are computer-generated.

  • Text: what is a 'usual programming language restriction'?

  • Syntax: there are potential ambiguities. E.g., with the character string "WEIGHT=1", a strict left-right parser would treat "WEIGHT" as NOMNAME and deduce invalid syntax, while another parser could treat "WEIGHT" as a key with value and find no NOMNAME. This should be resolved, e.g., by prohibiting predefined words as key identifiers.

  • Syntax/semantics: are integers and floating point values distinct?

  • Comments regarding ROOT: is this part of the standard, or is it an implementation detail that can be skipped? Is there an equivalence between strings in either direction of translation, e.g., if both formats appear as keys in the same weight set?

  • Semantics/interpretation: for some applications, events are certainly unweighted, for others they are not. Rejection sampling is a method that need not apply to future generators, e.g., machine-learning based sampling. I do not see these and similar scattered comments as relevant for the standard, so I would recommend to defer them to a separate section which may describe the current use inside the LHC software framework.

  • Semantics: The semantic equivalence between NOMINAL, DEFAULT, WEIGHT, 0 (or letter O?) and empty string appears arbitrary and redundant. It is easy to miss that only one should (or must?) appear, and confusing to any reader. I recommend that the authors decide on one of those and allow the others, if at all, as legacy variants that a reader but not a writer may support.

  • Usage: While it is clearly stated that the NOMINAL weight is special, I do not understand why this particular weight name cannot be annotated by key-value pairs. If all other weight names can be parsed and interpreted, it might be useful for an application to know which parameters apply to the nominal weight without having to look at metadata elsewhere.

  • Semantics: e.g., comments regarding IRREG: why not just state for any element whether it is mandatory or optional. Statements such as 'might not even exist' just confuse the reader.

  • Semantics: The text states that a setting such as "KEY=" without value may be interpreted as boolean true. To me, such a setting rather looks like an undefined value. Is this semantics actually useful?

  • Text: The terms GenEvent/GenRun are not defined. The statement where they appear is unclear to me, how should documentation be accessible?

  • Semantics: There is a paragraph regarding ordering of weight sets (vectors). This looks like an unfinished work item. I recommend to either state that this standard does cover weight sets, not just individual weights, or not. If yes, define sets either as ordered, partially ordered or unordered, and define the exact requirements on ordering so they can be assumed and respected by a software implementation. Analogous arguments apply to other paragraphs, such as the following section on weight manipulation.

  • Usage: there is a paragraph which hints at optimizations such as storing the keys separately and using integers instead in the event record. This may be seen as an implementation detail, but it should be clarified if and how such optimizations must be supported by a programming interface to this standard.

  • Flexibility: The values of 'weight' entries appear to be restricted to single floating-point values. If an event format allows other data types such as tables, complex numbers, etc. as individual 'weight' entries, would the proposed standard be sufficent, open to extensions, or would it become invalid?

Conclusion:

I understand that my remarks my be seen as pedantic and beyond the intentions of the document. However, all details and formalities do matter if somebody wants to produce a correctly working implementation, possibly without direct contact to the original authors or years later.

I believe that this standard and the ideas behind it are important and deserve publication and wide recognition. Unfortunately, in its current version I cannot recommend acceptance without substantial revision.

  • validity: -
  • significance: -
  • originality: -
  • clarity: -
  • formatting: -
  • grammar: -

Author:  Andy Buckley  on 2022-09-02  [id 2787]

(in reply to Report 1 on 2022-04-29)
Category:
answer to question
reply to objection

Thank you for the detailed consideration, though we think in some places there is a mismatch of expectation and intention for this document in the HEP MC event generation and user communities. We attempt to address all these points in the detailed responses below (response paragraphs beginning with a ">>") and in the updated version on arXiv. Our apologies that this update and response process has taken so long!

Comments and >> responses:

Major issues:

  1. The document is rather vague about its intended audience. It has been submitted by the MCnet community, and it reflects on upcoming simulation runs for LHC Run 3. It refers to software tools such as the HepMC event format that are currently employed by some part of the HEP community. If the intended audience was indeed restricted to this community and time scale, the proper form of the paper would be a public MCnet note; in this case I would not see justification for a journal publication.

We now specify the target audience as primarily authors of particle-level MC generators (which is broader than MCnet) and users of downstream codes and analyses using events produced following the standard. There is no identified end to the relevant time-period but of course we wish to encourage rapid uptake.

As regards the proper form of publication, there is no such concept as a “public MCnet note”: the standard form of publication of MC developments, both physics-oriented and technical, is a reviewed paper. The review process (as here) also assists in refining the content beyond what is possible within a small collaboration -- this perhaps highlights a distinction between large experiment collaborations within which internal review can be relatively exhaustive, and a small, loose community like MCnet. We note that the most obvious prior art in this area is the Les Houches Event Format paper Comput.Phys.Commun. 176 (2007) 300-304, which was peer reviewed and published in CPC, and has >500 citations. SciPost Codebases do not perform the same role; to our understanding SciPost Physics Core is the appropriate SciPost venue for papers on technical concepts related to physics studies.

Conversely, if the intended audience was larger, the authors should (1) state in concrete terms which API properties and version of currently existing software are relevant for the implementation of the intended standard, and (2) relate to a larger environment of Monte-Carlo simulation for current and future experiments, e.g., how it should apply to other existing or foreseen software frameworks, beyond the LHC, and to expected future developments such as NNLO-level simulation tools. I think that a standard for indexing MC event metadata could become very useful for the whole community.

We do not see that either the API calls nor specific software need to be specified. Ordered and named vectors of event weights have been supported in the HEPEVT and HepMC records for decades, which account for the vast majority of non-heavy-ion collider-physics event generation (and increasingly HepMC3 is being used in new HI studies). Exhaustively document additional formats and versions which do not support multi-weights or weight naming is beyond the scope of the paper, which is to define naming structures (i.e. string formats) and ordering conventions within that basic functionality, to enable better physics interoperability. NNLO and similar high-precision calculations do not change the nature of theory systematic variations as far as the authors can identify. The scope does not extend to completely generic MC-event metadata, but we note in passing the existence of a generic attribute mechanism in HepMC3, suitable for this task.

  1. Basic terms such as 'weight', 'event', 'particle', 'stream', 'vector', etc. must be defined at the beginning of the document in self-contained and formal terms as far as they are relevant. The technical definition must be abstracted from the physical interpretation. In the current text, they are introduced in a rather casual way, assuming working knowledge (e.g., with HepMC) that is not evident for users not familiar with 2022 LHC internals. The terms should be used consistently throughout the document.

We have introduced clearer definition of these terms. However, for the intended audience of both MC authors and users, these terms are mostly self-evident and we do not want to obstruct readability by over-formalising the context.

  1. The scope of the standard must be stated. I understand that the concrete syntax only defines the format of key identifiers in a lookup table. However, there are also various recommendations for the structure of this table (per event record), and there are remarks concerning the complete stream of records. This leaves any reader confused about which data objects are actually covered by the standard.

We cannot force generator authors to provide identical lists of run parameters, not least because the different models have different degrees of freedom. The scope of the standard is to provide a common structure within which such variations can be expressed and parsed by users, and to codify standard names for the rather small subset of parameters common to all (or nearly all) generator codes. This aspect is open to further extensions, and the multi-level block name structure allows custom variation parameters to nonetheless be identified as specific to a common physics aspect such as matrix elements or parton showers.

  1. It has to be stated whether and how the standard applies to data independent of their particular storage structure. At the end of the document there is a paragraph on optimization (i.e., using numeric indices and a global lookup table mixed with event-local keys). It is not obvious how such optimization interacts with the syntax and semantics introduced at the beginning, or whether it actually contradicts the normative part.

We have now clarified that this is motivation for the ordering standard, and is based on the de facto current usage. It does not influence the application of the naming standard, except in that the unphysical naming prefixes are exempt from this requirement as long as placed on the tail of the vector (as is stated explicitly in the following paragraph).

General remarks:

A data-format standard serves three purposes. A writer (human or computer) is enabled to construct a conforming data structure, a parser is enabled to correctly process the data syntax, and an interpreter is enabled to process the semantics as intended.

To this end, the standard definition must be written such that a human or computer can unambiguously decide whether a given data set is standard-conforming or not. The current draft is full of unclear terms like 'may', or 'should'; I do not see how it actually defines anything.

I recommend that the document clearly separates its formal part, namely the definitions of scope, terms, syntax and semantics, from informal additions. The formal part alone allows for writing a correct programming interface. The informal part may include examples, context, reasons, use cases, or best-practice recommendations. Actually, worked-out examples would be rather useful and could help to straighten the core of the document.

We have taken the advice to strengthen the language used for requirements as opposed to advice through the document. However, for the audience of generator authors and users, we feel a fully formalised spec would be a less effective form of communication: the EBNF notation already specifies the format fully for the purposes of a parser, while the semantic arguments for why this format evolved requires syntax. Worked-out examples are already provided: we are not sure what can be added to these.

Some details:

  • Sec. 2: The term 'particle-level' is misleading without context -- is the weight pertaining to a single electron? Since the detector simulation environment is not referred to in the standard, I recommend to use predefined technical data terms where possible.

“Particle-level” is the standard terminology for fully-exclusive collider events independent of detector effects, i.e. the output of shower+hadronisation event generators. We have now specified this in the introduction.

  • Syntax: it is not stated whether character case is meaningful or not, i.e., whether "NAME" is different from "Name". It is not stated whether keywords such as "NOMINAL" must be upper case.

The last sentence in the section on the “nominal/default weight” states that “Analysis tools should always perform a case-insensitive string comparison.” We moved this to just below the EBNF specification in order to clarify that this applies to the entire weight-name string.

(Actually, at the end of 2.2. there is a remark which suggests that the syntax might be case-insensitive, but this is difficult to spot, and it is again just a 'should' clause.) It might be necessary to specify the character encoding. This depends on whether the standard refers to abstract names or to their representation in storage, which is not evident in the present text.

We now specify that analysis tools must parse case-insensitively, and explain this is motivated/required by the established generator conventions (this is an agreement, not a diktat, so we are limited to what could be agreed). The character encoding need not be specified as we explicitly limit to the ASCII subset: this extra motivation is now made explicit.

  • Syntax: it is not clear without reading the text several times what the allowed characters for ALPHA actually are. To avoid all ambiguities, directly define ALPHA as a character set at the point where it appears. (Same for DIGIT).

Done.

  • Syntax: the 'usual standard form' of an integer or floating-point number is unclear. The writer of a parser, for instance, must exactly know which formats to allow and accept. Regarding floating-point, it must be stated what the allowed ranges are and whether numeric or textual reading/comparison is to be applied (cf. numbers differing in the 31st decimal), whether +0.0 is equal to -0.0, etc. All of this is relevant if the key strings themselves are computer-generated.

We’ve added an explicit definition of integer/floating points in the EBNF specification. However, what matters is that it be parseable by implementing languages. Comparisons will be done on the in-memory value, applying FP fuzziness as appropriate and according to the preferences of the code authors.

  • Text: what is a 'usual programming language restriction'?

The text specifies what this means; the reference to programming languages acts only as a justification for the adopted scheme.

  • Syntax: there are potential ambiguities. E.g., with the character string "WEIGHT=1", a strict left-right parser would treat "WEIGHT" as NOMNAME and deduce invalid syntax, while another parser could treat "WEIGHT" as a key with value and find no NOMNAME. This should be resolved, e.g., by prohibiting predefined words as key identifiers.

The KEY cannot be the same as NOMNAME. This constraint was indeed missing from the EBNF specification and we have added it now. Thanks for catching!

  • Syntax/semantics: are integers and floating point values distinct?

It doesn’t matter; the decision of what types to parse into (string/int/float) is the choice of the application wanting to parse and semantically identify the weight streams. Nevertheless, we have added a definition for the usual integer/floating point representation in EBNF to be explicit.

  • Comments regarding ROOT: is this part of the standard, or is it an implementation detail that can be skipped? Is there an equivalence between strings in either direction of translation, e.g., if both formats appear as keys in the same weight set?

It’s advice – we’ve clarified this in the text.

  • Semantics/interpretation: for some applications, events are certainly unweighted, for others they are not. Rejection sampling is a method that need not apply to future generators, e.g., machine-learning based sampling. I do not see these and similar scattered comments as relevant for the standard, so I would recommend to defer them to a separate section which may describe the current use inside the LHC software framework.

This is not the sort of weight being discussed here: the paper is specifically about variation weights, not weights from weighted generation of nominal events. The introduction now mentions this as part of the requested definition of weights, as well as in the section on default/nominal weights. Variation weights are multiplicative on the nominal generation/bias weights and remain relevant whatever the nominal-generation strategy.

  • Semantics: The semantic equivalence between NOMINAL, DEFAULT, WEIGHT, 0 (or letter O?) and empty string appears arbitrary and redundant. It is easy to miss that only one should (or must?) appear, and confusing to any reader. I recommend that the authors decide on one of those and allow the others, if at all, as legacy variants that a reader but not a writer may support.

We agree! But this format agreement expresses the compromises that were achievable given the prior art of established event generators with established conventions. They were adamant on retaining incompatible namings for the nominal stream! We have made it explicit in the text that the character “0” is the digit.

  • Usage: While it is clearly stated that the NOMINAL weight is special, I do not understand why this particular weight name cannot be annotated by key-value pairs. If all other weight names can be parsed and interpreted, it might be useful for an application to know which parameters apply to the nominal weight without having to look at metadata elsewhere.

It is important to be able to easily identify the nominal configuration with a simple name, also because such names are already well-established. Prior art exists for providing a variation-style stream for the nominal configuration, with its parameters written out.

  • Semantics: e.g., comments regarding IRREG: why not just state for any element whether it is mandatory or optional. Statements such as 'might not even exist' just confuse the reader.

There were two occurrences of this - one we’ve removed since it wasn’t really necessary to have (in the definition of IRREG-type weights), the other one is in the section on weight ordering and we think is necessary to motivate the strict requirement that IRREG-type weights really ought to be at the end of the weight vector. We think the wording is now clear.

  • Semantics: The text states that a setting such as "KEY=" without value may be interpreted as boolean true. To me, such a setting rather looks like an undefined value. Is this semantics actually useful?

A “BLOCK” is defined as: BLOCK = KEY , [ "=" , VALUE ] in EBNF notation, which allows for constructions like “KEY” and “KEY=VALUE” but not “KEY=”, since “,” is the concatenation operator, as opposed to the alternation operator “|”. We have clarified “value-assignment” in the text and made the boolean interpretation mandatory.

  • Text: The terms GenEvent/GenRun are not defined. The statement where they appear is unclear to me, how should documentation be accessible?

These are HepMC objects; the point is now made more generically with these as an explicit example.

  • Semantics: There is a paragraph regarding ordering of weight sets (vectors). This looks like an unfinished work item. I recommend to either state that this standard does cover weight sets, not just individual weights, or not. If yes, define sets either as ordered, partially ordered or unordered, and define the exact requirements on ordering so they can be assumed and respected by a software implementation. Analogous arguments apply to other paragraphs, such as the following section on weight manipulation.

The issue seems to be that these sections are quite discursive. We have made the concrete requirements stemming from the discussion more explicit and actionable. The introduction is now clearer that this paper addresses the ordering and the value-meanings of variation weights as well as stream-naming.

  • Usage: there is a paragraph which hints at optimizations such as storing the keys separately and using integers instead in the event record. This may be seen as an implementation detail, but it should be clarified if and how such optimizations must be supported by a programming interface to this standard.

This section has been clarified.

  • Flexibility: The values of 'weight' entries appear to be restricted to single floating-point values. If an event format allows other data types such as tables, complex numbers, etc. as individual 'weight' entries, would the proposed standard be sufficent, open to extensions, or would it become invalid?

There is no realistic prospect of this occurring, and it would be a huge feature-creep subject to abuse were we to introduce the possibility. Variation weights are defined as the ratio of (possibly subtracted) probability densities, which are single, real-valued, floating-point numbers. Where more structured information is needed, this is currently achieved via multiple streams requiring a combination recipe -- with the standard-proposal in this document largely motivated by making such recipes possible to automatically identify and apply.

Conclusion:

I understand that my remarks may be seen as pedantic and beyond the intentions of the document. However, all details and formalities do matter if somebody wants to produce a correctly working implementation, possibly without direct contact to the original authors or years later.

I believe that this standard and the ideas behind it are important and deserve publication and wide recognition. Unfortunately, in its current version I cannot recommend acceptance without substantial revision.

Understood, and we hope we have addressed the deficiencies and explained rationales where mismatches of expectation are intentional. We also note that this standard is already implemented in multiple generator and analysis codes, but that the paper remains important as a reference against which to check implementations and to explain the origins and limitations of the specified conventions and requirements.

---

## Round 2 · Referee Report · Anonymous (Referee 2) · 2022-5-2

Report

I think that in its previous form, submission is not a physics paper but technical note addressed to MC authors, may be only of MCnet community. I have not much to add to previous referee opinion. I agree with it.

On the other hand , standard for definition of weights used in Monte Carlo programs would be beneficial for the community. Technical side of such definition is present, but essential, physics side is missing. If it is not provided, document will be, in my opinion, nearly useless for the community of Monte Carlo users and another failed attempt for standarisation will take place.

If the authors can provide (and agree among themselves) on the section on ``limitation/(applicability domain) for the weight use'', paper can include necessary physics content and its publication may be then justified. My suggestion follow:

For each program it should be explained what are limitation in weights use, that is what is its applicability domain. Alternatively detailed reference to manuals with page numbers etc. can be given. This is not easy, as it touch approximations used in programs design. In my opinion that is essential for the readers. Finally, template of explanations (what is required from the authors for the weight to be used), would be beneficial for future programs (programs versions) documentation too.
  • validity: -
  • significance: -
  • originality: -
  • clarity: -
  • formatting: -
  • grammar: -

Author:  Andy Buckley  on 2022-09-02  [id 2788]

(in reply to Report 2 on 2022-05-02)

Our thanks for the comments. We have now uploaded a new version of the paper accounting for the many detailed comments from referee 1, cited here as supported by referee 2. We respond below to the more general remarks about the nature of this paper and what would make it publishable in SciPost Physics Core. We note that there is an established precedent for "community agreement" papers of this sort -- also extending to procedural collections of consensus such as https://inspirehep.net/literature?sort=mostrecent&size=25&page=1&q=find%20eprint%202003.07868 However, if there is a clearly more appropriate venue for this paper under the SciPost imprint, we are flexible! We respond "inline" below to the specific critiques:

I think that in its previous form, submission is not a physics paper but technical note addressed to MC authors, may be only of MCnet community. I have not much to add to previous referee opinion. I agree with it.

>> This is undoubtedly not just of interest to the MCnet community: it is relevant to the myriad of experiment and phenomenology users who need to form MC-systematic uncertainties from their multi-weighted event samples. The mess resulting from the absence of a standard has already led to proliferations of unscalable machinery in experiments to try and hide such details from physics users. This change to make the weights predictable and meaningful is relevant to all users of MC events as well as their producers.

On the other hand , standard for definition of weights used in Monte Carlo programs would be beneficial for the community. Technical side of such definition is present, but essential, physics side is missing. If it is not provided, document will be, in my opinion, nearly useless for the community of Monte Carlo users and another failed attempt for standarisation will take place.

>> We disagree, but have added some further physics context to the introduction, to explain typical applications of variation weights and the required combination rules. On the assertion that a standard introduced without a physics component will be a failure, we refer to the Les Houches Event Format paper (Comput.Phys.Commun. 176 (2007) 300-304, https://arxiv.org/abs/hep-ph/0609017, 510 citations), one of the most successful community standards in high-energy physics, whose text contains no reference whatsoever to physics applications! Out of the available SciPost journals, Physics Core seems the closest match to Comp Phys Commun.

If the authors can provide (and agree among themselves) on the section on limitation/(applicability domain) for the weight use'', paper can include necessary physics content and its publication may be then justified. My suggestion follow:

For each program it should be explained what are limitation in weights use, that is what is its applicability domain. Alternatively detailed reference to manuals with page numbers etc. can be given. This is not easy, as it touch approximations used in programs design. In my opinion that is essential for the readers. Finally, template of explanations (what is required from the authors for the weight to be used), would be beneficial for future programs (programs versions) documentation too.

>> We explicitly choose not to discuss issues of weight-applicability in this document: its purpose is not to advocate for weighting methods or to claim relevances that they do not have. We are specifying a convention for weight-stream organisation, to address issues in their already well-established domains of validity. We have extended the introduction to make clear some basic technical essentials for use of reweighting, such as the need for common support between the source and target distributions, but a context-specific discussion on what physics can and cannot be effectively unweighted would be boundless and off-topic.

With apologies, we do not understand the suggestion to refer to page numbers in manuals (what manuals?), the reference to approximations, or a “template of approximations”.

---

## Round 3 · Referee Report · Anonymous (Referee 3) · 2022-9-29

Strengths

1 - the paper provides a solution of an important standardization problem
2 - the solution is well thought out
3 - the solution is presented clearly (with minor exceptions)
4 - the solution is a consensus of the most important stake holders

Weaknesses

1 - there is a confusing phrase on page 6.

Report

As the authors describe very clearly, a practical and well defined standard for communicating the event level weight variations with input parameters from Monte Carlo event generators to analysis software packages is of crucial importance for making the most out of the experiments at the LHC. It is neither practical nor ecologically responsible to regenerate events for performing the many parameter scans that are performed in the analysis of LHC event samples.

By its nature, the paper does not present new physics results, but it should be published as an important reference for the community.

The proposed standard is practical and reflects the experience of the authors. The rationale for the design choices is explained well. The description of the standard is very clear (with one minor exception, see below) and can be adapted easily by authors and users of analysis software packages and by authors of other Monto Carlo event generators.

Nevertheless, I have suggest minor improvements:

In the last paragraph on page 6, the authors refer to "Non-standard weights or weights otherwise unknown to the parton shower". It is not explained why the parton shower component of event generators is singled out here. It is not clear to me whether this is an editorial oversight from an earlier draft or whether there is a reason for excluding other components (PDFs, ME, hadronization) here. The authors should consider dropping "or weights otherwise unknown to the parton shower" altogether or clarify their intentions.

Indeed, the final two paragraphs of section 2.3 are largely redundant with the preceding listing. The authors might want to consider merging the two.

Also in section 2.3, the authors might want to consider making the "USER" or "AUX" class marker obligatory for unblessed physical events.

Finally, reference [29] could be replaced by the publically available ISO/IEC 14977 standard.

Requested changes

1 - the authors should clarify why they single out the parton shower in the phrase "weights otherwise unknown to the parton shower" in the last paragraph on page 6.

  • validity: top
  • significance: high
  • originality: high
  • clarity: high
  • formatting: perfect
  • grammar: perfect

Author:  Andy Buckley  on 2022-10-03  [id 2870]

(in reply to Report 1 on 2022-09-29)
Category:
remark
answer to question

Our thanks for the positive feedback and for identifying the issue of duplication around the prefix-code options!

The duplication and confusing sentence in Section 2.3 have been resolved in a new version now submitted to arXiv to appear tomorrow. We have further explained the distinction between USER and no-prefix modes, and made it explicit that streams not for physical reweighting interpretation *must* be marked as such with a suitable prefix and vector position.

On the EBNF, while searching for the ISO reference we encountered the critique at https://dwheeler.com/essays/dont-use-iso-14977-ebnf.html and took the advice there to use the more compact, familiar, and accessible syntax variant by W3C. The new version cites the relevant W3C document.

---

## Round 3 · Author Response

Dear editor and referees,

Thank you for the helpful comments which have assisted us in alleviating ambiguities in the specification. And our apologies for the long delay in response time, due to some very heavy university commitments for both corresponding authors.

We have uploaded the improved version 3 to the arXiv, which we hope addresses the detailed points made by referee 1. As a general point we would like to highlight to both referees that there is no such object as a "MCnet public note" and that the journal review process has already added value in the intended way; by analogy with the very successful Les Houches Event Format paper, which is similar in many respects including a lack of detailed physics case or impact study, journal publication is established and appropriate for community standards such as this. Our understanding is that SciPost Physics Core is the appropriate venue for such papers which are not "codes or algorithms" cf. SciPost Physics Codebases, but we would be happy to be referred there if the relatively HEP-community aspect of this paper would fit its definition and processes better than Physics Core.

Best wishes,
Andy Buckley, for the editors

---

## Round 3 · List of Changes

All referee points have been considered and most accepted. The full set of responses will be posted to the referee comment, as the most efficient way to document the many detail changes.

---

## Round 4 · Referee Report · Anonymous (Referee 3) · 2022-10-11

Report

The revised version addresses all concerns in my previous report.

Section 2.3 is substantially improved. The present version makes clear when to use which prefixes.

Using the W3C reference for ENBF is fine. Also the grammar has become more concise by employing character classes.

---

## Round 4 · Referee Report · Anonymous (Referee 2) · 2022-10-16

Strengths

1) It covers important software application domain 2) It is clear

Weaknesses

1) It is missing section or at least statement about possible ambiguities

Report

I think that authors response missed my previous comments and my concerns remain valid.
Reweighting techniques are known and well established since many decades. They rely on mathematical
principles closely related to those of multidimensional integration variables changes.
To prevent biases, faulty applications, some rules need to be followed:
modified distribution need to be known and explicit. Usually it is not the case, approximations
and resulting ambiguities require evaluation. Who should take resposability: Monte Carlo authors, end users,
or somebody else?

Existence of ambiguity question, should be at least stated in the sumbmission. Clear and easy to spot warning
that somebody need to take responsability and do the hard work is needed. Good candidates would be
Monte Carlo authors who are the best to know necessary detail of the distributions to be modified by weights.
In general evaluation will depend on Monte Carlo and approximations used in it. Difficult ambiguity evaluation
can be delegated to users. This will substantially reduce size of user community.

Let me now go back to my previous concerns and author response.

-- I think that in its previous form, submission is not a physics paper but technical note addressed
-- to MC authors, may be only of MCnet community. I have not much to add to previous referee opinion.
-- I agree with it.

>> This is undoubtedly not just of interest to the MCnet community: it is relevant to the myriad of
>> experiment and phenomenology users who need to form MC-systematic uncertainties from their
>> multi-weighted event samples. The mess resulting from the absence of a standard has already led
>> to proliferations of unscalable machinery in experiments to try and hide such details from physics
>> users. This change to make the weights predictable and meaningful is relevant to all users of MC
>> events as well as their producers.

I agree, with they statement ``it is relevant to the myriad of experiment and phenomenology users'',
but to be useful physics and approximation details of the weight definitions must be given in submission
or it must be pointed where they can be found, or at least that user must look for the information.

-- On the other hand , standard for definition of weights used in Monte Carlo programs would be beneficial
-- for the community. Technical side of such definition is present, but essential, physics side is missing.
-- If it is not provided, document will be, in my opinion, nearly useless for the community of Monte Carlo
-- users and another failed attempt for standarisation will take place.

>> We disagree, but have added some further physics context to the introduction, to explain typical
>> applications of variation weights and the required combination rules. On the assertion that a standard
>> introduced without a physics component will be a failure, we refer to the Les Houches Event Format paper
>> (Comput.Phys.Commun. 176 (2007) 300-304, https://arxiv.org/abs/hep-ph/0609017, 510 citations), one of
>> the most successful community standards in high-energy physics, whose text contains no reference
>> whatsoever to physics applications! Out of the available SciPost journals, Physics Core seems the
>> closest match to Comp Phys Commun.

Usefulness of weights is not only about formats/standards/citations it is about their physics (and mathematical)
meaning. That is particular project dependent. It can not be just `examples' it is precondition for use.

-- If the authors can provide (and agree among themselves) on the section on limitation/(applicability
-- domain)for the weight use'', paper can include necessary physics content and its publication may be then
-- justified. My suggestion follow:

-- For each program it should be explained what are limitation in weights use, that is what is its
-- applicability domain. Alternatively detailed reference to manuals with page numbers etc. can be given.
-- This is not easy, as it touch approximations used in programs design. In my opinion that is essential
-- for the readers. Finally, template of explanations (what is required from the authors for the weight to
-- be used), would be beneficial for future programs (programs versions) documentation too.

>> We explicitly choose not to discuss issues of weight-applicability in this document: its purpose is not
>> to advocate for weighting methods or to claim relevances that they do not have. We are specifying a
>> convention for weight-stream organisation, to address issues in their already well-established domains
>> of validity. We have extended the introduction to make clear some basic technical essentials for use of
>> reweighting, such as the need for common support between the source and target distributions, but
>> a context-specific discussion on what physics can and cannot be effectively unweighted would be
>> boundless and off-topic.

I suspect, there is misunderstanding at this point. Weighting methods do not need explanation, they simply
reduce to redefinition of function which is to be integrated over massively multidimensional set of
variables and that is rather trivial from the point of view of mathematical
principles. My concern is, that to apply reweihgting techniques it is needed to understand what is the physics
content of orginal event sample and then of weights.

Side remark: Applications imprinting events into HepMC are plagued with standard stretching
(history entries hard process kinematic configurations different from the actual one, energy-momentum
non-conservation in vertices). In fact the stretching are understandable, needed, contribute to HepMC success,
but often confuse users. Your proposal undoubtedly will suffer from similar problems.

In my opinion, introduction of warning that validity (ambiguity evaluation) of re-weighting, require
details of assumptions/approximations used in event sample generation, is absolute minimum
before submission can be accepted. It must be stated that ambiguity is Monte Carlo, initialization variant
and user application dependent.

One may say that the topic is out of scope of the submission, but this nonetheless
need to be stated, at least. I am not asking for much. One or two sentences in summary and/or introduction
may be sufficient.

Requested changes

One may say that the topic is out of scope of the submission, but this nonetheless
need to be stated, at least. I am not asking for much. One or two sentences in summary and/or introduction
may be sufficient.

---

## Round 4 · Author Response

We have addressed the issues identified by the reviewer; our thanks for their efforts, which have again improved the presentation and fixed some ambiguities and minor errors.

---

## Round 4 · List of Changes

1. The duplication and confusing sentence in Section 2.3 have been resolved, and we have expanded on the meanings of different prefixes and explicitly that streams not for physical reweighting interpretation must be marked as such with a suitable prefix and vector position.

  2. Searching for the EBNF ISO reference highlighted that this form is only semi-openly available, and has been critiqued for some cryptic constructions that certainly affected the use-case here. We have taken the opportunity to switch to the more compact, familiar, and accessible variant by W3C, which is now cited.

  3. Minor corrections to the EBNF specification of allowed number formats.

---

## Editorial Decision

published